# Accounting for overdispersion in skilled antenatal care: Identifying determinants using Bangladesh Demographic and Health Survey 2022 data

Md. Muddasir Hossain Akib[1], Farhia Azrin[2]*, Bikash Pal[1], Md. Mustain Billah[1]

1 Department of Statistics, University of Dhaka, Dhaka, Bangladesh, 2 Institute of Statistical Research and Training, University of Dhaka, Dhaka, Bangladesh

* fazrin@isrt.ac.bd

## Abstract

One of the major components required to ensure safe motherhood is taking a sufficient number of skilled antenatal care (SANC) visits by women during pregnancy. Various social, demographic, and economic factors have a strong influence on it. According to the World Health Organization (WHO), four or more antenatal care (ANC) visits are required for uncomplicated pregnancies and delivery of a healthy baby. SANC serves this purpose efficiently. As a result, modeling the number of SANC visits and identifying its determinants is crucial for the Bangladesh government in adopting appropriate policies and programs. In this study, we used score tests to assess dispersion in the count data and selected the Generalized Poisson Regression Model (GPRM) based on the lowest AIC (15,839.2) and BIC (16,001.7) values. Analysis of BDHS 2022 data (n = 3,839) revealed that only 38.6% of women received the recommended four or more SANC visits, with an overall mean of 3.06 visits. Key predictors of higher SANC utilization included urban residence (IRR = 1.09, $p < 0.01$), higher women's education (IRR = 1.57, $p < 0.001$), exposure to media (IRR = 1.16, $p < 0.001$), and being in the richest wealth quintile (IRR = 1.22, $p < 0.001$). In contrast, women in the Sylhet region (IRR = 0.89, $p = 0.009$) and those with higher birth order had lower SANC visit rates.

## Introduction

Maternal mortality is a major concern in developing countries. Pregnant women may experience some complications that can lead to maternal and infant mortality. Globally, around 810 maternal deaths occur every day, mostly resulting from complications during pregnancy [1]. Furthermore, these pregnancy or childbirth related complications lead to 300,000 maternal deaths annually, the majority of these occurring in low-resource settings, which are largely preventable [2,3]. The highest number of

**Data availability statement:** The dataset utilized for this study is available in the Supporting Information.

**Funding:** The author(s) received no specific funding for this work.

**Competing interests:** The authors have declared that no competing interests exist.

deaths occurs in Southern Asia, accounting for 66%, followed by Sub-Saharan Africa (20%), which accounts for almost 86% of global maternal deaths [4]. It is evident that the most significant causes of death and disabilities occurring among women in Bangladesh are due to the complications of pregnancy and child delivery [5–7]. These preventable deaths during pregnancy and delivery period can be decreased with the aid of using the right care and scientific solutions provided by hospitals and clinics [8].

Antenatal care (ANC) is a set of necessary routine tests and counselling performed to train and prepare pregnant women for safe deliveries. ANC plays an important role in maternal health by offering numerous health functions, detects health problems and supports the well being of infants and mothers. Ensuring the health of mother and infant by the end of the pregnancy is considered the ultimate goal of ANC [9]. ANC-related healthcare has been sustained globally, originating nearly a century ago from models first developed in Europe [10]. The pregnant mothers become informed about possible pregnancy and delivery complications and their possible remedies throughout this medical supervision [11]. It is essential to access ANC services to lessen the risk of maternal and child deaths [12]. The World Health Organization (WHO) recommends that pregnant women take at least eight ANC examinations performed by skilled attendants to ensure safe motherhood during the whole period [13]. If a pregnant woman receives ANC-related services from professionally trained providers, then such healthcare-related services are defined as skilled ANC (SANC), as the providers elevate the required knowledge of safer motherhood-related practices and raise awareness among pregnant women to maintain healthy conditions throughout pregnancy [6]. However, according to available evidence, globally around 90% women utilize ANC by skilled providers at least one time, but only 60% of pregnant women received skilled ANC for at least four times, as recommended by WHO [14].

Bangladesh is a developing country, and the health status of this country has gradually improved over past few decades. However, progress in maternal health remains slow. It is evident that, the maternal mortality rate in Bangladesh has remained the same in past years with an estimated number of 196 maternal deaths per 100,000 live births which was almost similar in 2010 [15]. The target of sustainable development goals (SDGs) 3 regarding maternal mortality is to reduce the maternal mortality rate to fewer than 70 per 100,000 live births by 2030, which is difficult to achieve for Bangladesh [16]. As per the report of Bangladesh Health Facility Survey 2017 (BDHS 2017), almost 99% healthcare facilities in Bangladesh offer ANC services, but only 4% are adequately equipped to provide quality ANC services [17]. As a result, women are usually unable to receive the recommended ANC services during their pregnancy [18]. In the past, several studies conducted in Bangladesh revealed various factors to be directly associated with the antenatal healthcare of women during their pregnancy [19–22]. The available research on ANC mainly analyzed household-level data. It showed that region, type of place of residence, mother's age, mother's education, partner's education, the total number of children ever born, wealth index, and exposure to mass media have a significant impact on the pursuit of ANC-related healthcare [3,23–25].

The number of SANC visits is count data and usually Poisson regression model (PRM) is used to analyze this type of data [11,26]. However, real life count data are often overdispersed or underdispersed, as a result PRM becomes unsuitable [27]. By ignoring the dispersed nature of count data in modelling may provide misleading interpretations [28]. To the best of authors knowledge, no study had conducted any hypothesis testing on count data of SANC services of pregnant women to determine whether the data exhibit dispersion. As a result, this study aims to test some hypotheses to check the dispersion of SANC data. Following this, to compare different available models for handling dispersed count data and select the best-fitting model is another goal of this study. Finally, using the latest Bangladesh Demographic and Health Survey 2022 (BDHS 2022) data, the study wants to identify the most significant factors that influence skilled ANC visits among pregnant women in Bangladesh.

## Materials and methods

### Data and sampling design

For the purpose of analysis, in this paper, data have been extracted from the Bangladesh Demographic and Health Survey (BDHS), 2022–23. It is a nationally representative survey that is conducted as a part of the Demographic and Health Survey (DHS) program by the United States of America (USA). The data set is a retrospective survey. The information collection method is established on a two-stage stratified cluster sampling plan. At the initial stage, 675 enumeration areas were selected where EAs were selected with probability proportional to EA size. In the next and final step, 45 households per EA were systematically selected. All ever married women aged between 15–49 years who were either usual resident of the selected households or stayed in the selected households the night before the survey were eligible for the interview. After removing all the missing cases, we ended up with a dataset including the information of 3,839 respondents.

### Variables

In this study, the response variable of interest is the number of skilled antenatal care (SANC) visits received by women in Bangladesh during pregnancy. An ANC healthcare service is considered skilled if a pregnant woman receives care from one or more designated professionals such as, doctors, nurses, midwives, paramedics, family welfare visitors, sub-assistant community medical officers, or community skilled birth attendants [29]. Several socio-economic and demographic variables have been considered as explanatory variables in this study. The covariates namely are region (Dhaka, Barishal, Chittagong, Khulna, Mymensingh, Rajshahi, Rangpur, Sylhet) which are administrative divisions, type of residence (rural, urban), women's education level (no education, primary, secondary, higher), husband's education level (no education, primary, secondary, higher), women 's age in years (<20, 20–35, >35), wealth index (poor, middle, rich; derived using PCA based on ownership of household goods), exposure to media (exposed, unexposed), sex of household head (female, male), birth order (1, 2–3, 4 or higher), women 's religion (Muslim, others) and finally migration status (yes, no). As some variables and their corresponding categories were not directly available in the survey data, we have created and reclassified them. A pregnant woman is defined as being exposed to media if she reads newspapers or magazines, listens to the radio, or watches television at least once a week [20,21]. The variable "migration" was calculated based on respondents who had lived in their current residence for less than two years [11,30,31].

### Models

The data were analyzed by using bivariate and multivariate statistical methods. In bivariate analysis, chi square and ANOVA F test has been utilized to check the statistical significance of categorical and count dependent variable, respectively [32,33]. A variable was considered significant if the p-value appeared less than 0.05.

In practice, count data may show overdispersion, therefore, it is important to consider its statistical significance. Three versions of the score test statistics are available to check the statistical significance of overdispersion.

Dean and Lawless (1989) [34]:

$$z_i = \frac{((y_i - \mu_i^2) - y_i)}{\sqrt{(2\mu_i)}}.$$

Cameron and Trivedi (1990) [35]:

$$z_i = \frac{((y_i - \mu_i^2) - y_i)}{\mu_i}.$$

Winkelmann (2008) [36]:

$$z_i = \frac{((y_i - \mu_i^2) - y_i)}{2\mu_i}.$$

We initially considered four different regression models: Poisson, negative binomial, generalized Poisson and Conway-Maxwell-Poisson [37]. The Poisson regression parametric model is the benchmark for count data [38]. Count data demonstrates the number of times an occasion took place and it cannot be demonstrated by a classical linear regression due to violation of the assumptions of continuity and normally dispersed dependent variable [39]. The regression model uses the log link, and the model is linear in terms of the logarithm of the count data and assumes equidispersion. Therefore, we applied the Poisson regression model in a generalized modeling framework to analyze the number of SANC visits taken by women during pregnancy in Bangladesh [40]. The probability mass function of Poisson regression is as follows:

$$f\left(Y_i = y_i | x_i\right) = \frac{e^{-ui} \times \mu^{yi}}{y_i!} ; y_i = 0, 1, 2, \ldots, \ where \ \mu_i = e^{Xi,\beta o}.$$

Although in real life scenario, mostly the data are overdispersed, meaning the mean and variance are not equal [41]. Due to this, other options are usually considered to fit the model. Mostly, Negative binomial model is usually used to deal the situation of overdispersion, which is a gamma mixing distribution for the mean parameter [42,43]. The response variable $Y$ follows negative binomial $(\mu_i, \tau)$ distribution with the following probability mass function:

$$f\left(Y_i = y_i; \mu_i, \tau\right) = \frac{\Gamma\left(y_i + \frac{1}{\tau}\right)}{y_i!\Gamma\left(\frac{1}{\tau}\right)} \left(1 + \tau\mu_i\right)^{-\frac{1}{\tau}} \left(1 + \frac{1}{\tau\mu_i}\right)^{-y_i} ; y_i = 0, 1, 2, \ldots$$

with $E\left(Y_i\right) = \mu_i$ and $Var\left(Y_i\right) = \mu_i\left(1 + \tau\mu_i\right)$, $\tau > 0$. The extra variation due to overdispersion is $\left(1 + \tau\mu_i\right)$ that depends on the mean $\mu_i$. Another available option for modelling count data is Generalized Poisson (GP) which is able to deal with both overdispersion and underdispersion, although the range for underdispersion is limited [44,45].

By considering some parametric transformation, the following probability mass function is followed by GP [46].

$$f\left(y_i, \mu_i, \alpha\right) = \left(\frac{\mu_i}{1 + \alpha\mu_i}\right)^{y_i} \frac{\left(1 + \alpha y_i\right)}{y_i!} \exp\left[\frac{-\mu_i\left(1 + \alpha y_i\right)}{1 + \alpha\mu_i}\right] ; y_i = 0, 1, 2, \ldots$$

with $E\left(Y_i\right) = \mu_i$ and $Var\left(Y_i\right) = \mu_i\left(1 + \alpha\mu_i\right)^2$. If $\alpha = 0$, the GP turns into Poisson, for $\alpha > 0$ and $\alpha < 0$, it reflects overdispersion and underdispersion respectively. Furthermore, another suitable option to model count data is Conway-Maxwell

Poisson (CoMPois) which was first presented by Conway for modeling queuing system [47]. This can deal with both underdispersion and overdispersion [48,49]. The probability mass function of CoMPois model is

$$P\left(Y_i = y_i \middle| \lambda, v\right) = \frac{\lambda^v}{(y_i!)^v Z(\lambda, v)}, \ y = 0, 1, 2, \ldots, n, \ \text{where } z(\lambda, v) = \sum_{s=0}^{\infty} \frac{\lambda^s}{(s!)^v}, \ \lambda > 1, \ v \geq 0$$

with $E\left(Y_i\right) = \mu_i \approx \lambda^{1/v} - \frac{v-1}{2v}$ and $\text{Var}\left(Y_i\right) \approx \frac{1}{v}\lambda^{1/v}$. Here, $z(\lambda, v)$ is the normalization constant, $v > 1$ indicates underdispersion and $v < 1$ indicates overdispersion. This model is theoretically useful in many aspects, as it is a part of the family of two parameter power series distributions. Additionally, various probability mass functions can be modelled by this such as geometric distribution ($v = 0$), Poisson distribution ($v = 1$) and Bernoulli distribution ($\lambda < 1$ and $v \rightarrow \infty$). However, standard Conway-Maxwell Poisson distribution cannot model mean directly which is a major drawback. Therefore, we considered mean-parameterized Conway-Maxwell Poisson regression to make the interpretation simple as this model allows to model the mean counts directly [50]. The model can be used by maximizing the log-likelihood $\log L\left(\theta, v\right) = \sum_{i=1}^{n} \left(y_i \log\lambda\left(\mu_i, v\right) - v \log\left(y_i!\right) - \log z\left(\lambda\left(\mu_i, v\right), v\right)\right)$ [50]. Here the link function is $E\left(Y_i\right) = \mu_i = \exp(\mathbf{x}_i^\top \boldsymbol{\beta})$, where the vector of explanatory variables are $\mathbf{x}_i = (x_{i1}, \ldots, x_{ij}, ..x_{ip})'$ and the vector of regression coefficients are $\boldsymbol{\beta} = \left(\beta_{i1}, \ldots, \beta_{ij}, ..\beta_{ip}\right)'$ and the parameters $\boldsymbol{\theta} = \left(\boldsymbol{\beta}, v\right)'$. For the sake of this study, the mpcmp package in R is utilized to estimate the model parameters using Newton-Raphson iteration technique [51].

To choose the most suitable model among these available options Akaike Information Criteria (AIC) and Bayesian Information Criteria (BIC) is utilized. The model with the smallest AIC and BIC is considered the best fitted model [52]. For a model with $p$ parameters and $n$ observations, AIC and BIC can be represented as, $\text{AIC} = -2l\left(\hat{\theta}, \mathbf{y}\right) + 2p$, $\text{BIC} = -2l\left(\hat{\theta}, \mathbf{y}\right) + p \log(n)$, where $l\left(\hat{\theta}, \mathbf{y}\right)$ represents the log-likelihood. All the statistical analyses were performed using IBM SPSS Statistics version 27 and R software version 4.3.3.

## Results

Table 1 represents the summary statistics of SANC visits in various socio-economic and demographic characteristics where to explore the overall scenario, frequency of 4 or more SANC visits and mean of frequency of SANC visit was considered. According to the table, type of residence, women's education level, women's age at last birth, husband's education level, wealth index, birth order, media exposure, religion and migration status are significantly associated with receiving four or more SANC visits (Yes, No) determined by chi-square test. These factors are also correlated with frequency of SANC visits along with region, as determined by ANOVA. It is visible that, overall, only 38.6% of women receive the recommended four or more SANC with a mean of 3.06. The analysis reflected that women's educational qualification, age of women, husband's educational qualification, economic condition of family, birth order, exposure to media, religion, migration status is significantly associated with the frequency of SANC visits. Women living in Dhaka had a rate of 40.4% of SANC utilization while in Sylhet the rate was only 30.3%. Women with higher educational status were having a higher proportion (61.7%) of receiving proper SANC visits compared to women with other educational status and the scenario is similar when considering mean of SANC visits where women with higher education had a mean of 4.4. Coverage of proper SANC visits was higher (51.2%) in urban women while the rate was only 32.4% in rural women. Women aged between 20–35 years and more than 35 years received 40.0% and 39.0% recommended number of SANC by WHO, respectively, where only 33.9% young women utilized this service. Women who were at highest wealth index were at a higher rate (54.8%) to receive proper SANC, on the other hand, only 24.7% and 34.5% women in poor and middle wealth index received this service. Media exposure also plays a significant role where women who are exposed to media had a higher frequency (47.2%) and mean (3.5) of SANC visit. Furthermore, women who migrated reflected a higher proportion (46.2%) of at least 4 SANC visits while 37.7% non-migrated women utilized this service.

**Table 1. Mean and percentage of SANC visits by the socio-economic and demographic variables with p-values.**

| Characteristics | Number of women (%) | % of women who had $\geq 4$ SANC visits | p-value[a] | Mean of frequency of SANC visits | p-value[b] |
|---|---|---|---|---|---|
| Total | 3839(100.0) | 38.6 | | 3.06 | |
| **Region** | | | 0.051 | | **<0.001** |
| Dhaka | 586(15.3) | 40.4 | | 3.37 | |
| Barishal | 422(11.0) | 33.2 | | 2.90 | |
| Chittagong | 641(16.7) | 38.8 | | 3.08 | |
| Khulna | 431(11.2) | 44.3 | | 3.45 | |
| Mymensingh | 471(12.3) | 45.0 | | 2.92 | |
| Rajshahi | 381(9.9) | 40.7 | | 3.13 | |
| Rangpur | 451(11.7) | 35.5 | | 2.89 | |
| Sylhet | 456(11.9) | 30.3 | | 2.63 | |
| **Type of residence** | | | **0.006** | | **<0.001** |
| Rural | 2571(67.0) | 32.4 | | 2.74 | |
| Urban | 1268(33.0) | 51.2 | | 3.70 | |
| **Women's education level** | | | **<0.001** | | **<0.001** |
| No Education | 200(5.2) | 23.0 | | 1.86 | |
| Primary | 941(23.8) | 24.2 | | 2.20 | |
| Secondary | 2028(52.8) | 38.7 | | 3.09 | |
| Higher | 697(18.2) | 61.7 | | 4.44 | |
| **Women's age at last birth** | | | **<0.001** | | **0.004** |
| <20 | 853(22.2) | 33.9 | | 2.83 | |
| 20-35 | 2783(72.6) | 40.0 | | 3.12 | |
| >35 | 200(5.2) | 39.0 | | 3.13 | |
| **Husband's education level** | | | **<0.001** | | **<0.001** |
| No Education | 573(14.9) | 26.2 | | 2.20 | |
| Primary | 1186(30.9) | 26.8 | | 2.41 | |
| Secondary | 1341(34.9) | 41.0 | | 3.23 | |
| Higher | 739(19.2) | 62.8 | | 4.45 | |
| **Wealth index** | | | **<0.001** | | **<0.001** |
| Middle | 768(20.0) | 34.5 | | 2.85 | |
| Poor | 1550(40.4) | 24.7 | | 2.21 | |
| Rich | 1521(39.6) | 54.8 | | 4.02 | |
| **Birth order** | | | **<0.001** | | **<0.001** |
| 1 | 1348(35.1) | 42.1 | | 3.30 | |
| 2–3 | 2104(54.8) | 38.5 | | 3.04 | |
| 4 or higher | 387(10.1) | 26.9 | | 2.31 | |
| **Media exposure** | | | **<0.001** | | **<0.001** |
| Unexposed | 1685(43.9) | 27.7 | | 2.45 | |
| Exposed | 2154(56.1) | 47.2 | | 3.53 | |
| **Religion** | | | **<0.001** | | **<0.001** |
| Others | 342(8.9) | 47.7 | | 3.58 | |
| Muslim | 3497(91.1) | 37.7 | | 3.01 | |
| **Sex of household head** | | | 0.186 | | **0.014** |
| Female | 401(10.4) | 41.6 | | 3.33 | |
| Male | 3438(89.6) | 38.2 | | 3.03 | |
| **Migration status** | | | **0.001** | | **<0.001** |
| No | 3426(89.2) | 37.7 | | 3.00 | |
| Yes | 413(10.8) | 46.2 | | 3.54 | |

[a]p-values are based on the Chi-square test for testing variation among the proportions

[b]p-values are based on the Analysis of Variance (ANOVA) and F-test for testing the variation among the means.

In Table 2, three distinct dispersion tests are done to check the overdispersion of Poisson model. All of these models revealed significance, which indicates the presence of overdispersion in the SANC count data.

Table 3 represents the Akaike Information Criterion (AIC) and Bayesian Information Criterion (BIC) to assess the model performance of four different models – PRM, GPRM, NBRM, and CMPRM which are used to fit count data. The AIC values stand for 16135, 15839, 15929, and 15880, respectively, for PRM, GPRM, NBRM, and CMPRM models. Additionally, the BIC values for PRM, GPRM, NBRM, and CMPRM models are 16291, 16002, 16091, and 16043, respectively. From both AIC and BIC metric, it is visible that the GPRM model fits the data better in comparison to the other models. The minimum AIC and BIC values for GPRM concludes it as the superior model and due to this, the results produced from the GPRM model have been prioritized for further discussion in this study.

Table 4 illustrates the factors that are influencing the usage of SANC. According to the table, region, place of residence, educational qualification of women, age of women at last birth, educational qualification of husband, economical position in society, birth order, exposure to mass media, religion, sex of family head and migration status have a significant impact on the usage of SANC. The administrative division played a crucial role where the rate of SANC utilization was 11% lower (IRR: 0.89, 95% CI: 0.82–0.97, p<0.01) in Sylhet compared to Dhaka. The type of place of residence plays an important role in terms of receiving proper SANC visits. The rate of SANC visit is 9% higher (IRR: 1.09, 95% CI: 1.04–1.15, p<0.001) in urban women compared to rural women. The educational qualification of women is significantly impacting the rate of SANC visit. Women with primary, secondary, and higher education have 22% (IRR: 1.22, 95% CI: 1.07–1.41, p<0.01), 44% (IRR: 1.44, 95% CI: 1.25–1.65, p<0.001) and 57% (IRR: 1.57, 95% CI: 1.35–1.82, p<0.001) higher rate of receiving SANC, respectively, compared to women with no education. Young women are significantly less likely to receive SANC as the rate of receiving SANC is 13% (IRR: 1.13, 95% CI: 1.06–1.20, p<0.001) and 30% (IRR: 1.30, 95% CI: 1.16–1.47, p<0.001) higher in women aged between 20–35 years and more than 35 years, respectively, compared to women aged less than 20 years. Women who were economically having a lower condition in society were 14% (IRR: 0.86, 95% CI: 0.81–0.92, p<0.001) less likely to receive SANC compared to middle class women. On the other hand, Women belonging to higher economic class were significantly more likely to receive SANC, with a 22% increased likelihood (IRR: 1.22, 95% CI: 1.15–1.30, p<0.001). Birth order is negatively associated with the rate of receiving SANC. Having birth order of 2 or 3 and 4 or higher had 9% (IRR: 0.91, 95% CI: 0.86–0.96, p<0.01) and 20% (IRR: 0.80, 95% CI: 0.73–0.89, p<0.001) ower rates of receiving SANC compared to the first birth order. Exposure to the media also plays a vital role. Women exposed to mass media had a 16% (IRR: 1.16, 95% CI: 1.10–1.21, p<0.001) higher rate compared to unexposed women. Migration is very closely associated with the rate of SANC visits where migrated women had a 7% (IRR: 1.07, 95% CI: 1.00–1.14, p<0.1) higher rate of SANC usage compared to unmigrated women.

**Table 2. Detection of overdispersion based on score tests of significance of overdispersion to SANC count data in Bangladesh.**

| Test | Z-Score | Standard Error (SE) | p-value |
|---|---|---|---|
| Dean & Lawless | −1.8 | 2.1 | <0.10 |
| Cameron & Trivedi | −2.8 | 1.1 | <0.01 |
| Winkelmann | −2.8 | 0.5 | <0.01 |

**Table 3. Selection of model based on AIC and BIC for SANC count data in Bangladesh.**

| SANC | AIC | | | | BIC | | | |
|---|---|---|---|---|---|---|---|---|
| | PRM | GPRM | NBRM | CMPRM | PRM | GPRM | NBRM | CMPRM |
| | 16134.8 | 15839.2 | 15928.5 | 15880.6 | 16291.1 | 16001.7 | 16091.1 | 16043.2 |

**Table 4. Effect of the selected covariates on the SANC visits along with p-values and IRRs with 95% CI, obtained from the Generalized Poisson Regression Model (GPRM).**

| Covariate | Estimate | SE | p-value | IRR | 95% CI (IRR) | |
|---|---|---|---|---|---|---|
| | | | | | Lower | Upper |
| **Region** | | | | | | |
| Dhaka | | | | | | |
| Barishal | −0.000 | 0.044 | 0.998 | 1.00 | 0.92 | 1.09 |
| Chittagong | 0.034 | 0.038 | 0.374 | 1.03 | 0.96 | 1.11 |
| Khulna | 0.034 | 0.041 | 0.411 | 1.03 | 0.95 | 1.12 |
| Mymensingh | 0.069 | 0.042 | 0.100 | 1.07 | 0.99 | 1.16 |
| Rajshahi | −0.017 | 0.044 | 0.688 | 0.98 | 0.90 | 1.07 |
| Rangpur | −0.050 | 0.044 | 0.258 | 0.95 | 0.87 | 1.04 |
| Sylhet | −0.116 | 0.044 | **0.009** | 0.89 | 0.82 | 0.97 |
| **Type of residence** | | | | | | |
| Rural | | | | | | |
| Urban | 0.090 | 0.024 | **<0.001** | 1.09 | 1.04 | 1.15 |
| **Women's education level** | | | | | | |
| No Education | | | | | | |
| Primary | 0.203 | 0.071 | **0.004** | 1.22 | 1.07 | 1.41 |
| Secondary | 0.362 | 0.070 | **<0.001** | 1.44 | 1.25 | 1.65 |
| Higher | 0.452 | 0.076 | **<0.001** | 1.57 | 1.35 | 1.82 |
| **Women's age at last birth** | | | | | | |
| <20 | | | | | | |
| 20-35 | 0.120 | 0.033 | **<0.001** | 1.13 | 1.06 | 1.20 |
| >35 | 0.265 | 0.060 | **<0.001** | 1.30 | 1.16 | 1.47 |
| **Husband's education level** | | | | | | |
| No Education | | | | | | |
| Primary | −0.007 | 0.042 | 0.868 | 0.99 | 0.91 | 1.08 |
| Secondary | 0.115 | 0.043 | **0.007** | 1.12 | 1.03 | 1.22 |
| Higher | 0.267 | 0.049 | **<0.001** | 1.31 | 1.19 | 1.44 |
| **Wealth index** | | | | | | |
| Middle | | | | | | |
| Poor | −0.145 | 0.034 | **<0.001** | 0.86 | 0.81 | 0.92 |
| Rich | 0.198 | 0.031 | **<0.001** | 1.22 | 1.15 | 1.30 |
| **Birth order** | | | | | | |
| 1 | | | | | | |
| 2–3 | −0.097 | 0.028 | **0.001** | 0.91 | 0.86 | 0.96 |
| 4 or higher | −0.217 | 0.051 | **<0.001** | 0.80 | 0.73 | 0.89 |
| **Media exposure** | | | | | | |
| Unexposed | | | | | | |
| Exposed | 0.144 | 0.025 | **<0.001** | 1.16 | 1.10 | 1.21 |
| **Religion** | | | | | | |
| Others | | | | | | |
| Muslim | −0.098 | 0.037 | **0.007** | 0.91 | 0.84 | 0.97 |
| **Sex of household head** | | | | | | |
| Female | | | | | | |
| Male | −0.059 | 0.035 | 0.091 | 0.94 | 0.88 | 1.01 |
| **Migration status** | | | | | | |
| No | | | | | | |
| Yes | 0.065 | 0.034 | 0.058 | 1.07 | 1.00 | 1.14 |

## Discussion

This study utilized the latest nationally representative dataset (BDHS 2022) from Bangladesh to examine the dispersion of SANC visit data, identify the most appropriate statistical model, and determine the key factors influencing skilled ANC utilization. The findings reflect that the SANC data is truly dispersed in nature, as a result, models that account for dispersion provide a better fit. Additionally, predisposing factors such as age, educational qualification of both women and their husband were significantly associated with skilled ANC utilization. Moreover, enabling factors such as place of residence, wealth status, exposure to mass media also showed significant associations with SANC. Furthermore, migration status was found to influence the frequency of SANC visits. To our knowledge, no prior study has conducted any hypothesis testing for overdispersion on count data of SANC in Bangladesh. This study fills the gap by applying three well established statistical tests to verify the presence of dispersion in the data. Particularly from Cameron & Trivedi and Winkelmann tests, it is strongly evident that the data is overdispersed. As a result, alternative count data models were applied to model the data for accurate inference. Among all other suitable models, the generalized Poisson regression model (GPRM) showed the lowest AIC and BIC values. Therefore, this study has used GPRM to model the data to find the factors that impact the utilization of SANC.

Age impacts the exposure towards utilizing SANC service. This study found that younger women were less likely to utilize this service and this finding is in line with previous research [53]. A study conducted over East African countries investigated that, older women are aware regarding the importance of receiving skilled ANC care [54]. According to WHO, social stigma and low education limits care-seeking among younger mothers [55]. Additionally, shyness, lack of family support, misconception regarding ANC care, fear of medical procedures may contribute to lower utilization of SANC among young mothers.

The findings of this study show that women with education are more prone to receive SANC service from skilled providers compared to mothers with no education, which is consistent with the findings of other studies from different parts of the world [24,53,56–60]. The possible explanation for this scenario might be the awareness and knowledge of women regarding their health which comes with education.

Wealth status is a major determining factor to receive SANC. According to the findings of the study, women in the higher wealth quintile are more likely to receive SANC, consistent with previous research findings [53,56,58,60,61]. Affordability of medical cost, easier access to medical services and knowledge about the importance of ANC service might be probable explanation for this scenario. Women in the lowest wealth quintile had the lowest coverage of SANC, which may reflect persistent structural and financial barriers, despite the presence of free or subsidized services through public and NGO-supported programs.

Women residing in rural areas were less likely to utilize SANC compared to urban women. This finding is consistent with former research [60,62]. A study found that, walking long distance to reach health facilities, insufficient midwives, rude behavior of healthcare providers are contributing factor less utilization of ANC service among rural mothers [63]. Additionally, financial difficulties, cultural beliefs, lack of awareness, and less availability of quality hospitals might be some others factors to influence rural mothers in receiving less SANC.

Furthermore, this study found that exposure to media significantly impacts the uptake of SANC which is parallel with previous research [24,53,56,64,65]. Increased awareness, empowerment through information, positive health messaging may influence this pattern. They have better access to information about the importance of ANC compared to unexposed mothers which encourages care-seeking behavior [65].

Migration status also impacts the frequency of SANC visits. Migrated women are more likely to utilize SANC visits compared to non-migrated women. Usually, people migrate from rural to urban areas and this may have an impact on utilizing the service. The possible explanation for this scenario might be easy access to healthcare, supportive social networks and potential positive influence from the surrounding environment.

These findings reflect the importance for simple and targeted actions to help young, uneducated, rural, and poor women get skilled ANC care. Improving health centers in rural areas, spreading awareness through media, and giving

financial help to poor families can make a big difference. Community support programs can also encourage more women to seek care. Working together across sectors is key to ensuring all women can access good maternal health services.

## Conclusion

This study highlights significant socio-demographic and economic determinants influencing the number of skilled ante-natal care (SANC) visits among women in Bangladesh, using the most recent BDHS 2022 data. Statistical tests confirmed overdispersion in the count data, and the Generalized Poisson Regression Model provided the best fit. Key factors positively associated with higher SANC utilization include maternal education, urban residence, media exposure, wealth status, and migration. Findings underscore the need for targeted interventions, especially among rural, less-educated, and economically disadvantaged women, to improve SANC coverage and ultimately reduce maternal and neonatal health risks. We recommend expanding community-based outreach programs in underserved regions and enhancing media campaigns focused on maternal health education. Additionally, increasing the availability of trained health providers in rural areas and integrating ANC awareness into school curricula can further boost service utilization and safe motherhood outcomes.

## Limitations

This study has several limitations. First, the analysis is based on cross-sectional data from the BDHS 2022, which limits the ability to establish causal relationships between predictors and SANC visits. Second, some important factors such as cultural beliefs, quality of care, and availability of health facilities could not be included due to data limitations. Third, recall bias may affect responses related to ANC visits, especially among women reporting events from several months prior. Lastly, while the study accounts for multiple socio-demographic variables, potential unobserved confounders may still influence the results.

## Supporting information

**S1 File. Study dataset.**
(CSV)

## Acknowledgments

The authors would like to express their sincere gratitude to the Demographic and Health Surveys (DHS) Program for granting access to the BDHS 2022 dataset used in this study.

## Author contributions

**Conceptualization:** Md. Muddasir Hossain Akib.

**Formal analysis:** Md. Muddasir Hossain Akib, Farhia Azrin, Md. Mustain Billah.

**Writing – original draft:** Md. Muddasir Hossain Akib, Farhia Azrin.

**Writing – review & editing:** Bikash Pal.

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
