## [Decision Letter · Decision Letter 0]

25 Jul 2025

Dear Dr. Azrin,

Thank you for submitting your manuscript to PLOS ONE. After careful consideration, we feel that it has merit but does not fully meet PLOS ONE’s publication criteria as it currently stands. Therefore, we invite you to submit a revised version of the manuscript that addresses the points raised during the review process.

We look forward to receiving your revised manuscript.

Kind regards,

Lufuno Makhado, Ph.D.

Academic Editor

PLOS ONE

Journal Requirements:

Additional Editor Comments :

Thank you for submitting your manuscript titled "Accounting for Overdispersion in Skilled Antenatal Care: Identifying Determinants Using Bangladesh Demographic and Health Survey 2022 Data." This topic is both timely and highly relevant to maternal health policy in Bangladesh and similar contexts. Your use of Generalized Poisson Regression to address overdispersion is well-supported and methodologically sound.

To enhance your manuscript and work towards publication, it is important to address several key issues raised by the reviewers. First, clarity in integrating your statistical findings into the narrative is essential, especially within the Results and Discussion sections. Additionally, it is crucial to report p-values, confidence intervals, and incidence rate ratios (IRRs) directly in the text rather than solely in the tables. For instance, when discussing urban women's higher utilization of Skilled Antenatal Care (SANC), it would be beneficial to specify that “urban residence is associated with a 9% increase in SANC visits (IRR = 1.09, p < 0.001).”

The reviewers also raised relevant questions about the lower SANC coverage among women in the middle-income category compared to both poor and wealthy groups. It would be helpful to explore possible structural or contextual factors that could explain this phenomenon, such as the potential lack of NGO support or financial barriers that might be affecting these women.

Furthermore, expanding on the policy implications of your findings is essential, particularly in terms of targeting interventions aimed at underserved groups like middle-income women, rural women, and those with low media exposure. It is important to ensure that your recommendations are directly linked to your statistical results to strengthen their relevance.

In addition, revising the manuscript for clarity and coherence will significantly improve its quality. Some sections may contain awkward phrasing or repetition, especially in the introduction and conclusion. You might consider seeking assistance from a native English speaker or a professional editing service to enhance the manuscript's readability.

Lastly, it is vital to ensure that all numerical findings in the tables, particularly in Tables 1 and 4, are adequately reflected and interpreted in the text, rather than simply referenced.

I appreciate the effort you've invested in this research and look forward to receiving your revised manuscript that addresses these points.

Reviewers' comments:

Reviewer's Responses to Questions

**Comments to the Author**

1. Is the manuscript technically sound, and do the data support the conclusions?

Reviewer #1: Yes

Reviewer #2: Yes

2. Has the statistical analysis been performed appropriately and rigorously?

Reviewer #1: Yes

Reviewer #2: Yes

3. Have the authors made all data underlying the findings in their manuscript fully available?

Reviewer #1: Yes

Reviewer #2: Yes

4. Is the manuscript presented in an intelligible fashion and written in standard English?

Reviewer #1: Yes

Reviewer #2: Yes

Reviewer #1: The article is well written with minor technical errors, however, there are gaps identified in data analysis and discussion of findings as explained in the attached report.TITLE: ACCOUNTING FOR OVERDISPERSION IN SKILLED ANTENATAL CARE: IDENTIFYING DETERMINANTS USING BANGLADESH DEMOGRAPHIC AND HEALTH SURVEY 2022 DATA

PRESENTATION OF THE RESULTS

1. To Validate Observations Statistically:

• Your summary shows differences in SANC (Skilled Antenatal Care) visits across education, location, age, and wealth index.

• Statistical tests (e.g., chi-square, ANOVA, regression) help determine if these differences are statistically significant or just due to chance.

2. To Identify Key Predictors:

• Understanding which factors (education, wealth, age, etc.) most strongly influence proper SANC visits can guide policy and intervention.

3. To Control for Confounding:

• For example, urban women may also be more educated. A multivariate analysis (like logistic regression) helps isolate the independent effect of each variable.

4. To Improve Clarity and Credibility:

• Presenting statistical relationships (with p-values, confidence intervals, etc.) strengthens the scientific rigor of your findings.

DISCUSSION OF RESULTS

Please improve the discussion by the following aspects

Statistical Support:

Include whether the differences in SANC uptake across wealth groups are statistically significant.

Mention any tests used (e.g., chi-square, logistic regression) and their results (e.g., p-values, odds ratios).

Clarify the Middle-Income Paradox:

Consider exploring why middle-income women might be underserved:

Are they excluded from free services?

Do they face barriers like cost or distance without the support that poorer women get?

Use Data to Strengthen Claims:

Refer to actual percentages or mean values from your study to support each point.

For example: “Women in the highest wealth quintile had a 54.8% SANC coverage, compared to 24.7% in the poorest and 34.5% in the middle-income group.”

Policy Implications:

Suggest what this means for health policy: Should outreach be expanded to middle-income women? Should subsidies be adjusted?

Reviewer #2: The methodology used is suitable for the study and data was analysed and presented well. However, the author should make amendments as highlighted on the manuscript. Mostly the amendments are on word usage and sentence construction.

.

Reviewer #1: No

Reviewer #2: No

---

## [Author Response · Author response to Decision Letter 1]

19 Aug 2025

Authors’ responses to the points raised by the Academic Editor and reviewers

First of all, we would like to thank the Academic Editor and the reviewers for reviewing our joint work entitled “Accounting for overdispersion in skilled antenatal care: Identifying determinants using Bangladesh Demographic and Health Survey 2022 data” and providing their valuable suggestions about our manuscript. All the suggestions raised by the Academic Editor and reviewers have been addressed in the revised version which are categorically pointed out in the followings.

Responses to Reviewer 1:

PRESENTATION OF THE RESULTS:

1. To Validate Observations Statistically:

Your summary shows differences in SANC (Skilled Antenatal Care) visits across education, location, age, and wealth index. Statistical tests (e.g., chi-square, ANOVA, regression) help determine if these differences are statistically significant or just due to chance.

Response: Thank you for your valuable suggestion. We agree that statistical tests are essential for identifying significant differences across the various groups considered in the analysis. In Table 1, we have already conducted chi-square and ANOVA tests to assess the statistical significance of these differences. Specifically, for categorical response variable (Yes/No), we used the chi-square test, and for continuous response (number of ANC visits), we performed ANOVA. The corresponding p-values are reported in the table. Additionally, in Table 4, we performed a Generalized Poisson Regression to further evaluate statistical significance where the p-values are also available.

2. To Identify Key Predictors:

Understanding which factors (education, wealth, age, etc.) most strongly influence proper SANC visits can guide policy and intervention.

Response: Thank you for your insightful comment. To highlight the most important factors influencing SANC visits, we have revised the results section by emphasizing these key factors. Additionally, we have updated the discussion with suggestions for policy and intervention.

3. To Control for Confounding:

For example, urban women may also be more educated. A multivariate analysis (like logistic regression) helps isolate the independent effect of each variable.

Response: Thank you so much for your insightful suggestion. We agree that controlling for confounding is important to better understand the independent effect of each variable. As our outcome variable is count data, we have already conducted a multivariate regression using Generalized Poisson Regression, as shown in Table 4. This allows us to account for potential confounders, such as the possibility that urban women may also be more educated and better understand the contribution of each factor.

4. To Improve Clarity and Credibility:

Presenting statistical relationships (with p-values, confidence intervals, etc.) strengthens the scientific rigor of your findings.

Response: Thank you for your thoughtful suggestion. We agree that presenting statistical relationships, such as p-values and confidence intervals enhances the scientific rigor of the findings. To address this, we have added the confidence intervals for the Incidence Rate Ratios (IRRs) in Table 4, the p-values were already included in the table. Additionally, we also have updated the writings of the result by adding p-values and confidence intervals.

DISCUSSION OF RESULTS:

1. Statistical Support:

Include whether the differences in SANC uptake across wealth groups are statistically significant. Mention any tests used (e.g., chi-square, logistic regression) and their results (e.g., p-values, odds ratios).

Response: Thank you for your valuable comment. The statistical significance across wealth groups are mentioned in result section and Table 1 and Table 4. To make it clearer, we have revised the writings of the result. Including these again in the discussion may cause repetition. Nevertheless, we appreciate your insight, and if you feel it would improve clarity, we are willing to add the percentages to the discussion section.

2. Clarify the Middle-Income Paradox:

Consider exploring why middle-income women might be underserved: Are they excluded from free services? Do they face barriers like cost or distance without the support that poorer women get?

Response: Thank you for pointing this out. We reviewed the manuscript and noticed that we mistakenly mentioned the middle-income group as receiving less skilled antenatal care, which is not what the results actually showed. We're sorry for this error. We have now corrected the statement and made sure the discussion reflects the findings more clearly. We’ve also explained the possible reasons behind the middle-income group’s situation without misrepresenting the data.

3. Use Data to Strengthen Claims:

Refer to actual percentages or mean values from your study to support each point.

For example: “Women in the highest wealth quintile had a 54.8% SANC coverage compared to 24.7% in the poorest and 34.5% in the middle-income group.”

Response: Thank you for this valuable suggestion. The exact percentages are already presented in the results section, and we felt that including them again in the discussion might lead to repetition. However, we truly appreciate your perspective, and if you believe it would enhance clarity, we are happy to incorporate the percentages into the discussion as well.

4. Policy Implications:

Suggest what this means for health policy: Should outreach be expanded to middle-income women? Should subsidies be adjusted?

Response: Thank you for your helpful comment. Since we corrected the earlier mistake about the middle-income group, we also updated the discussion. We now clearly reflect what the results show. We have added a point suggesting that policymakers may still look at whether outreach and subsidies are reaching all groups fairly, including middle income women.

Responses to Reviewer 2:

The methodology used is suitable for the study and data was analysed and presented well. However, the author should make amendments as highlighted on the manuscript. Mostly the amendments are on word usage and sentence construction.

Response: Thank you for your valuable feedback. We appreciate your positive comments regarding the suitability of the methodology and the data presentation. We have carefully reviewed the manuscript and addressed all the suggested amendments, particularly those related to word usage and sentence construction. We believe these changes have improved the clarity and overall quality of the manuscript.

Response to Editor:

Thank you for submitting your manuscript titled "Accounting for Overdispersion in Skilled Antenatal Care: Identifying Determinants Using Bangladesh Demographic and Health Survey 2022 Data." This topic is both timely and highly relevant to maternal health policy in Bangladesh and similar contexts. Your use of Generalized Poisson Regression to address overdispersion is well-supported and methodologically sound.

To enhance your manuscript and work towards publication, it is important to address several key issues raised by the reviewers. First, clarity in integrating your statistical findings into the narrative is essential, especially within the Results and Discussion sections. Additionally, it is crucial to report p-values, confidence intervals, and incidence rate ratios (IRRs) directly in the text rather than solely in the tables. For instance, when discussing urban women's higher utilization of Skilled Antenatal Care (SANC), it would be beneficial to specify that “urban residence is associated with a 9% increase in SANC visits (IRR = 1.09, p < 0.001).”

The reviewers also raised relevant questions about the lower SANC coverage among women in the middle-income category compared to both poor and wealthy groups. It would be helpful to explore possible structural or contextual factors that could explain this phenomenon, such as the potential lack of NGO support or financial barriers that might be affecting these women.

Furthermore, expanding on the policy implications of your findings is essential, particularly in terms of targeting interventions aimed at underserved groups like middle-income women, rural women, and those with low media exposure. It is important to ensure that your recommendations are directly linked to your statistical results to strengthen their relevance.

In addition, revising the manuscript for clarity and coherence will significantly improve its quality. Some sections may contain awkward phrasing or repetition, especially in the introduction and conclusion. You might consider seeking assistance from a native English speaker or a professional editing service to enhance the manuscript's readability.

Lastly, it is vital to ensure that all numerical findings in the tables, particularly in Tables 1 and 4, are adequately reflected and interpreted in the text, rather than simply referenced.

I appreciate the effort you've invested in this research and look forward to receiving your revised manuscript that addresses these points.

Response: We appreciate your insightful feedback. In response, we have revised the manuscript to clearly integrate statistical results (IRRs, confidence intervals, and p-values) into the narrative, especially in the results section. We also corrected an earlier mistake where we mentioned that middle-income women had lower SANC coverage. This has now been fixed to reflect the actual findings. The discussion has been updated to explore the situation of middle-income women more accurately, including possible financial and structural barriers they may still face. Policy implications are now more clearly linked to our findings, highlighting underserved groups such as rural and middle-income women. We have also improved the clarity and flow of the manuscript through language edits and made sure that key numerical results from Tables 1 and 4 are interpreted directly in the text.

---

## [Decision Letter · Decision Letter 1]

30 Dec 2025

Dear Dr. Azrin,

plosone@plos.org. . . . A letter that responds to each point raised by the academic editor and reviewer(s). You should upload this letter as a separate file labeled 'Response to Reviewers'.A marked-up copy of your manuscript that highlights changes made to the original version. You should upload this as a separate file labeled 'Revised Manuscript with Track Changes'.An unmarked version of your revised paper without tracked changes. You should upload this as a separate file labeled 'Manuscript'.

We look forward to receiving your revised manuscript.

Kind regards,

Dr Syed Khurram Azmat, PhD, MPH, MD

Academic Editor

PLOS One

Journal Requirements:

**Additional Editor Comments:**

The authors are requested to respond to ALL feedback from the peer review.

Reviewers' comments:

Reviewer's Responses to Questions

**Comments to the Author**

Reviewer #2: (No Response)

2. Is the manuscript technically sound, and do the data support the conclusions?

Reviewer #2: Yes

3. Has the statistical analysis been performed appropriately and rigorously?

Reviewer #2: Yes

4. Have the authors made all data underlying the findings in their manuscript fully available?

Reviewer #2: Yes

5. Is the manuscript presented in an intelligible fashion and written in standard English?

Reviewer #2: Yes

Reviewer #2: the author addressed some of the comments, while other comments were not addressed. The highlighted text in the manuscript was not addressed. Can the comments be addressed, or reasons for not addressing them be provided?

.

Reviewer #2: No

---

## [Author Response · Author response to Decision Letter 2]

23 Feb 2026

We have addressed all the reviewer comments to the best of our ability, and where certain comments could not be addressed, we have provided explanations in the manuscript and the response document. Regarding the highlighted text mentioned, we carefully checked the manuscript but did not find any highlighted sections.

---

## [Editor Report · Decision Letter 2]

17 Mar 2026

Accounting for overdispersion in skilled antenatal care: Identifying determinants using Bangladesh Demographic and Health Survey 2022 data

PONE-D-25-34622R2

Dear Dr. Azrin,

We’re pleased to inform you that your manuscript has been judged scientifically suitable for publication and will be formally accepted for publication once it meets all outstanding technical requirements.

Kind regards,

Syed Khurram Azmat, PhD, MPH, MD

Academic Editor

PLOS One
---

## [Editor Report · Acceptance letter]

PONE-D-25-34622R2

PLOS One

Dear Dr. Azrin,

I'm pleased to inform you that your manuscript has been deemed suitable for publication in PLOS One. Congratulations! Your manuscript is now being handed over to our production team.

Kind regards,

on behalf of

Dr. Syed Khurram Azmat

Academic Editor

PLOS One